# A comprehensive study of CYP2E1 and its role in carcass characteristics and chemical lamb meat quality in different Indonesian sheep breeds

Ratna Sholatia Harahap[1,2☯], Asep Gunawan[3☯], Yuni Cahya Endrawati[3], Huda Shalahudin Darusman[4,5], Göran Andersson[6], Ronny Rachman Noor[3]*

**1** Faculty of Animal Science, Post-Doctoral Animal Production and Technology Student, IPB University, Bogor, Indonesia, **2** Faculty of Animal Science, Jambi University, Jambi, Indonesia, **3** Faculty of Animal Science, Department of Animal Production and Technology, IPB University, Bogor, Indonesia, **4** Department of Anatomy, Physiology and Pharmacology, School of Veterinary and Biomedical Sciences, IPB University, Bogor, Indonesia, **5** Primate Research Centre, Institute of Research and Community Service IPB University, Bogor, Indonesia, **6** Department of Animal Biosciences, Swedish University of Agriculture Sciences, Uppsala, Sweden

☯ These authors contributed equally to this work.
* ronny_noor@apps.ipb.ac.id

## Abstract

The role of CYP2E1 in oxidation is essential for its effects on meat quality. This study used 200 Indonesian sheep (*Ovis aries*) to determine the SNP g allele frequencies. g. 50658168 T>C of *CYP2E1* gene located in 3´-UTR region and their genetic association with lamb quality traits, including carcass characteristics, retail cut carcass, physicochemical lamb, fatty acid, cholesterol, flavor and odor, and mineral content. Further, the level of *CYP2E1* mRNA and CYP2E1 protein expression in muscle were determined and correlated with lamb quality traits. *CYP2E1* gene polymorphisms were identified using Polymerase Chain Reaction-Restriction Fragment Length Polymorphism (PCR-RFLP) analysis. The *CYP2E1* mRNA expression levels in phenotypically divergent sheep populations were analyzed using Quantitative Real Time-PCR (qRT-PCR). Immunohistochemistry (IHC) and hematoxylin-eosin (HE) staining analysis used three samples each in the high and low lamb quality groups based on pH value and tenderness. An association study of *CYP2E1* gene polymorphisms was performed using General Linear Model (GLM) analysis. The genetic association between the CC, CT, and TT genotypes at the SNP g. 50658168 T>C *CYP2E1* gene and lamb quality traits were significant (P<0.05), including carcass characteristics, retail cut carcass, fatty acid, cholesterol, flavor, and odor. Lambs with the CT genotype had a higher mRNA and protein expression in high lamb quality traits. The highest CYP2E1 protein expression was localized in the longissimus dorsi. The group sample with high lamb quality had a higher area and perimeter of muscle cells. *CYP2E1* can be used as a genetic marker for selecting sheep with high meat quality.

**Data Availability Statement:** All relevant data are within the manuscript and its Supporting Information files.

**Funding:** -AG and RSH received an award from The Directorate General of Resources for Science, Technology, and Higher Education, Ministry of Research, Technology, and Higher Education, with grant number: 077/SP2H/LT/DRPM/2021, dated March 18, 2021 - RRN and RSH received an award from Post-Doctoral Project 2024 IPB University with grant number: 3/IT3.D12/SP/DAPT.PD/2024, date February 28, 2024. -The funders had no role in the study design, data collection, analysis, publication decisions, or manuscript preparation.

**Competing interests:** The authors have declared that no competing interests exist.

# Introduction

High protein contents of both animal and plant proteins are essential for growth and development and, thus, vital components in a healthy diet [1]. The current international Recommended Dietary Allowance (RDA) for protein is 0.8 g/kg BW (body weight)/day, regardless of age [2]. Proteins from plants have lower anabolic properties compared with animal proteins due to lower levels of essential amino acids such as leucine and lysine. This lower level is because plant-based proteins have less of an anabolic effect than animal proteins. Animal-based protein diets are critical for maintaining skeletal muscle mass [3]. Therefore, consuming good-quality animal proteins can enhance muscle protein synthesis rate and support muscle mass more than plant proteins [4,5]. FAO data [6] shows that plant protein consumption in Indonesia is higher than animal protein (65.70% or 40.77 grams/capita/year vs. 34.30% or 21.29 grams/capita/year). However, the consumption of animal protein in this community has increased yearly, whereas plant protein consumption has decreased. The total consumption of animal protein in 2015 was 31.59% or 18.50 grams/capita/year, while plant protein consumption was 68.41% or 40.06 grams/capita/year [7]. Based on these data and the fact that animal-based protein diets are critical for the maintenance of muscle mass among the elderly, it is warranted that food products of animal origin need to be increased.

In Indonesia, the supply and demand for meat are dominated by chicken (66%) and beef (21%) [8]. Beef is a red meat, apart from pork and lamb. Meeting the need for meat in Indonesia, especially beef, depends on imports. So, food consumption diversification (FCD) from meat must help meet these needs. With Indonesia being a Muslim-majority country, Halal food influences the buying decisions of many Indonesian consumers. Lamb is one of the preferred choices for realizing an FCD program. Lamb is the second most preferred meat after beef, and consumers are most willing to pay more for lamb and sheep meat because of its taste, tenderness, and perceived high animal welfare attributes [9]. Indonesia's sheep meat imports have increased enormously recently, with an average annual growth rate of 17% [8].

Nevertheless, lamb is frequently underrated as a culinary product because of its stereotype. Consumers' unwillingness to consume mutton is likely due to excessive fattiness, commonly associated with inferior sensory attributes [10]. The consumption of lambs is considered to cause degenerative diseases [11,12]. High intake of saturated fatty acids (SFAs) based on increased consumption of lamb meat is a significant factor leading to an increased risk of lamb consumption diseases, which can increase blood cholesterol levels, as has been well established, that contribute to atherosclerotic growth [12,13]. The intake of trans fatty acids is associated with an increased risk of all-cause mortality and cardiovascular diseases [14]. In addition, consumers said that the aversion of customers to so-called "mutton flavor" is virtually insignificant if the content of back fat and intramuscular fat (marbling) is reduced [12]. Therefore, lamb meat quality must be improved by producing high-quality sheep with body weight still young to slaughter, called balibu (less than five months old) and batibu (weaning, less than three months old).

Currently, the genetic quality of sheep is mainly improved by classical breeding methods. Breeding programs are supported by genomic selection and various molecular genetic tools to increase the rates of genetic gain for the production traits of interest. Meat quality traits can be improved using informative molecular markers with significant and positive effects on these attributes in marker-assisted selection [15]. One of the genes we studied is the *CYP2E1* (Cytochrome P450 Family 2 Subfamily E Member 1) gene. The *CYP2E1* gene is located on sheep chromosome 22 and has nine exons (NC_019479.2). Single Nucleotide Polymorphisms (SNPs) in *CYP2E1* significantly affect meat quality, that is, pH, tenderness, and fatty acid content. For example, the SNP g.50657948 T>G of *CYP2E1* showed that the GG genotype was

associated with high-quality lamb meat traits [16]. As a member of the oxide reductase cytochrome family, the CYP2E1 enzyme oxidizes a variety of small molecule endogenous substrates, such as cholesterol biosynthesis and cholesterol conversion to bile acids, xenobiotics, ethanol, formation of steroid hormones, androgens, and estrogens, metabolism of vitamin D3 to the active 1,25-dihydroxy vitamin D3, omega hydroxylation of fatty acids, and biotransformation of exogenous xenobiotics [17,18].

The essential role of CYP2E1 in these oxidation processes is a fundamental aspect of how this gene affects meat quality. Skeletal muscle fibers are classified according to their main metabolic and contractile properties [19]. Metabolic fiber types consist of oxidative (indicated by the red color that is rich in myoglobin, which is the oxygen carrier and pigment responsible for the red color) and glycolytic (indicated by the white color, which is almost devoid of myoglobin due to minimal requirements of oxygen) [20,21]. Therefore, understanding how CYP2E1 protein expression affects lamb meat quality and skeletal muscle tissue characteristics is essential. Our study was conducted to determine the SNP g. 50658168 T>C of *CYP2E1* gene and its association with lamb quality traits, including carcass characteristics, retail cut carcass, physical properties of lamb, fatty acids, cholesterol, flavor and odor, and mineral content, which was studied comprehensively and for the first time we also determined the nature of protein expression of CYP2E1 in muscle by IHC and HE staining.

## Materials and methods

The animal procedures were approved by the Local Ethics Committee for Animal Experimentation at the Institutional Animal Care and Use Committee (IACUC) issued by IPB University (approval no 117–2018 IPB), Indonesia.

### Animals, housing, and feeding

The samples used in this study were collected from 200 rams consisting of 10 Barbados cross sheep (BCS), 10 Compass Agrinac sheep (CAS), 10 Garut composite sheep (GCS), 18 Javanese fat-tail sheep (JFTS), 20 Garut sheep (GS), 27 Jonggol sheep (JS), and 105 Javanese thin-tail sheep (JTTS). The BCS, CAS, and GCS were crossbreed sheep taken from the Center for Research and Development of Animal Husbandry, Bogor, Indonesia (Latitude/Longitude: 6° 35'06.6"S 106°48'24.4" E). BCS was crossbred from local Sumatera sheep (50%) and Barbados black belly sheep (50%). CAS was crossbred from the local Sumatera (50%), St. Croix sheep (25%), and Barbados black belly sheep (25%). GCS was crossbred with indigenous Garut sheep (50%), St. Croix sheep (25%), and Moulton Charolais sheep (25%). IPB University developed JS sheep in the Jonggol Animal Science Teaching and Research Unit (JASTRU) (6°28'24.3″ S, 107°00'49.7″ E). The JFTS, JTTS, and GS are local breeds and samples were obtained from several farms in West Java, Indonesia. For the association study analysis, we used 100 rams from the JTTS consisting of CC (n = 33), CT (n = 58), and TT (n = 9). All rams were housed in group cages and given king grass forage (*Pennisetum purpureophoides*) ad libitum and supplemented with GT-03 Indofeed concentrate in the amount of 400 g/head/day in the afternoon, while water was given ad libitum.

### Slaughter procedure, traits description, and sample collection

The slaughter of 100 ram lambs was carried out based on the guidelines of the Indonesian performance test for halal slaughtering methods and under the supervision of Local Ethics Committee for Animal Experimentation at the Institutional Animal Care and Use Committee (IACUC) issued by IPB. Briefly, all rams were slaughtered at 10–12 months of age with a body

weight ranging between 20–30 kg at a commercial abattoir, PT Pramana Pangan Utama (PPU) Slaughter House.

The ram's slaughter was carried out on healthy or uninjured animals. The slaughtering process consists of pre-slaughter and slaughter. Pre-slaughter includes handling and restricting sheep and giving them ad libitum feed and water the day before slaughter. This procedure was done to minimize stress before the slaughtering process. The rams are rested in a pen far enough from the slaughtering process to alleviate stress and fear. The weight of the ram lambs was measured one hour before slaughter (final weight). The entire sample collection and slaughtering process was carried out under the supervision of a veterinarian using analgesia and anesthesia methods. The sacrifice process used a sharp blade coupled with a smooth incision in a single cut through the carotid artery and jugular vein. The rams were bleeding profusely and were handled until they became unconscious. The rams were hung so that the blood was completely drained from the body, then skinned and eviscerated, and the non-carcass components were separated, such as blood, skin, head, and feet, were weighed.

Carcass weight was measured immediately after skinning (hot carcass weight) and after chilling (4°C for 24 h) (cold carcass weight). The dressing percentage was calculated as the ratio of the cold carcass weight to the pre-slaughter weight multiplied by 100. The carcasses were split along the vertebral column into two halves. The right half of the carcass was divided into eight pieces based on commercial cuts (neck, shank, shoulder, breast, rack, loin, flank, and leg). These parts are divided into meat, bone, and fat. All meat cuts were weighed separately. Longissimus dorsi (LD) samples were collected for genomic DNA purification and lamb quality analyses (fatty acid, mineral content, flavor odor, cholesterol, protein expression: IHC and HE staining). In addition, the liver tissues were taken for mRNA expression analysis, while the *biceps femoris* samples were used to determine the physicochemical properties of lamb. All samples were placed on ice and stored at -20°C.

## Lamb quality analysis

Our study conducted a meticulous analysis of a variety of lamb quality traits, including carcass characteristics, retail cut composition, physical properties of lamb, fatty acid, cholesterol, flavor and odor, and mineral content. This comprehensive lamb quality analysis, performed on 100 slaughtered sheep samples, provides valuable insights into the intricate relationship between *CYP2E1* gene polymorphism and lamb quality.

## Physicochemical composition of lamb

The *biceps femoris* muscle from the right half of the carcass was used for the lamb's physical properties, including pH, tenderness, cooking loss, and water holding capacity. Meat pH was measured using a pH meter after the carcass had wilted for 24 h postmortem (ultimate pH). Two pieces of meat were weighed before being boiled for cooking loss and tenderness analysis. The meat was pierced with a bimetal thermometer and boiled in a water bath until its internal temperature reached 75–80°C. The meat was removed and left to rest at room temperature until it reached a constant weight. A piece of meat was weighed to obtain weight loss during cooking [22], while the other pieces were used for tenderness analysis. Cooked meat samples are shaped according to standards in the form of a cylinder with a diameter of 1.27 cm in the direction of the meat fibers, 3–5 cm long. The sample is then subjected to a slicing knife on the tool crosswise until it is split. The level of meat tenderness was measured in the *biceps femoris* muscle by measuring the amount of force (kg/cm$^2$) required to cut the meat's core, as indicated by the Warner Bratzler Shear Force (WBSF) needle. The greater the force needed to cut

the meat's core, the lower (more complex) its tenderness. The water-binding capacity was measured using the press method approach proposed by Hamm [23].

## Fatty acid composition of lamb

LD muscle from the right half of the carcass was used for fatty acid (FA) composition analysis. FA composition was analyzed using gas chromatography (GC), as described by the AOAC 969.333 extraction method [24]. Gas Chromatograms obtained on an RTX-WAX capillary column from polyethylene glycol (30 m × 0.25 mm × 0.25 μm) with a temperature range of 20˚C to 250˚C (Restek Co., Bellefonte, USA, cat. 12423). LD muscle was extracted using a chloroform-methanol solvent (2:1 v/v) to obtain the fat fraction. FA methyl esters were prepared by mixing saponified fat with boron trifluoride and then dissolving it in hexane. An Agilent gas chromatography system (6890N, Agilent Technologies, Santa Clara, CA, USA) was used to determine the FA composition. FA was identified by comparing the retention times of the methyl esters of the FA samples to those of the standards (FAME Mix, Sigma, Burlington, MA, USA, cat. 189–19). Concentrations were expressed relative to the amount of meat in mg/100 g (%). The total FA includes fat content, saturated fatty acids (SFA), unsaturated fatty acids (UFA), monounsaturated fatty acids (MUFA), and polyunsaturated fatty acids (PUFA) [25].

## Cholesterol content

The cholesterol content was analyzed using saponification and measurement steps. Saponification: About two grams of fat obtained from each sample was saponified using the method described by Vanderplanck et al. [26]. Cholesterol measurements were analyzed using gas chromatography-mass spectrometry (GC-MS). Gas Chromatograms obtained on an RTX-WAX capillary column from polyethylene glycol (30 m × 0.25 mm × 0.25 μm) (Restek Co., Bellefonte, PA, USA, cat. 12423). Cholesterol identification was performed by co-chromatography, and sample retention times were compared with standard retention times (Sigma, Burlington, MA, USA, ®C8667). Cholesterol content was expressed as mg/100 g of the LD samples.

## Flavor and odor measurements

LD samples (500 g) were used for flavor and odor analyses performed using methods described by Listyarini et al. [27]. Volatile odor and flavor compounds were extracted using the Likens-Nickerson method, which combines distillation and extraction with a solvent simultaneously using a gas chromatography-mass spectrometry (GC-MS) tool (GC-MS Agilent 7890A, 5975C XL EI/CI). Gas Chromatograms obtained on a HP-5MS capillary column from 5% Phenyl-Methylpolysiloxane (30 m × 250 μm × 0.25 μm) (Agilent Technologies, Santa Clara, CA, USA). The measured parameters were 4-methyloctanoic acid (MOA), 4-ethyloctanoic acid (EOA), 4-methylnonanoic acid (MNA), 3-methylindole (MI), 3-methylphenol (MP), and 4-methyl phenol (MP).

## Mineral content analysis

The mineral content of the LD muscle was measured using the IK LP-04.10-LT-1.0 method according to the AOAC (2015) Official Method 969.08 (AOAC 4.8.02). Mineral content analysis was carried out using the atomic absorption spectrometry (AAS) method, which uses light absorption to determine the concentration of specific metal atoms in a solid or liquid by evaporating the sample in a flame. The analyzed parameters included iron (Fe), potassium (K), selenium (Se), and zinc (Zn).

## Polymorphism determination of *CYP2E1* gene

Our research was conducted with the utmost precision and care. We used two hundred rams to identify *CYP2E1* gene polymorphism using a highly accurate method-PCR-RFLP (Polymerase Chain Reaction-Restriction Fragment Length Polymorphism). The genomic DNA was purified using a Genomic DNA Mini Kit (Geneaid Biotech, New Taipei, Taiwan) following the manufacturer's protocol, ensuring the reliability and accuracy of our results. SNP were obtained from the RNA Sequencing (RNA-Seq) results and the new SNP g. 50658168 T>C of the *CYP2E1* gene located in 3'-UTR regions was used in this study. The pair of primers with the amplification product of the 426 bp target DNA (F:5'- CCT TGA CTT CCC TGT CAG TA -3' and R: 5' - CAG AGC TTC AAG CCA AAA GG - 3') to amplify the *CYP2E1* gene based on the sheep reference genome sequence (NCBI accession NC_019479.2). PCR amplification using the AB System machine (Hyland Scientific, Silvana, WA, USA) with the initial processes was pre-denatured for 1 min at 95˚C. The next stage was denaturation for 15 s at 95˚C, annealing for 15 s at 60˚C, and elongation for 10 s at 72˚C for 35 cycles. The final step was elongation for 1 min at 72˚C. The PCR amplification results were detected using 1.5% agarose gel electrophoresis. The obtained *CYP2E1* gene PCR products were, furthermore, genotyped using digestion with the *Aci*I restriction enzyme (CCGC restriction sites) by PCR-RFLP for four h at 37˚C. The digested product was separated on a 2% agarose gel, and the fragments were visualized using a UV Transilluminator (Alpha Imager, Alpha Innotech, Santa Clara, CA, USA). The genotype's *CYP2E1* gene consisted of TT: 426 bp, CT: 426, 358, 68 bp, and CC: 68 bp and 358 bp.

## Quantitative real-time PCR analysis of *CYP2E1* gene

The quantitative real-time PCR analysis was carried out to observe the *CYP2E1* gene expression based on differences in genotype. Liver tissue samples for gene expression analysis were selected from a total of 100 rams (CC (n = 33), CT (n = 58), and TT (n = 9)). A total of 9 samples were selected representing three genotypes, CC, CT, and TT, each genotype consisting of three individuals. Total RNA from liver tissue samples obtained from nine different Javanese thin-tail and Garut composite sheep rams was isolated using the RNeasy Mini Kit (Qiagen, Hilden, Germany). Three groups of rams were formed based on the TT, CT, and CC genotypes with high and low pH and tenderness values. First-strand complementary DNA (cDNA) was synthesized from individual RNA samples using a First Strand cDNA Transcriptor Synthesis kit (Thermo Fisher Scientific, Vilnius, Lithuania). cDNA amplification was performed using the specific primers listed in Table 1, derived from the ovine *CYP2E1* sequence, and designed using MEGA 7.0. Each run examined the cDNA sample and the no-template control in a

**Table 1. SNP *CYP2E1* gene information and primer sequences.**

| Gene name | Accession number | Primer sequence | Application | TA (˚C) | Size of PCR | Restriction Enzyme | SNP | Product genotyping (bp) |
|---|---|---|---|---|---|---|---|---|
| *CYP2E1* | NC_019479.2 | >F: 5'- CCT TGA CTT CCC TGT CAG TA -'3<br>>R: 5'- CAG AGC TTC AAG CCA AAA GG -'3 | Genotyping | 60 | 426 bp | *Aci*I (CCGC) | g. 50658168 T>C | TT: 426 bp<br>CT: 68, 358, and 426 bp<br>CC: 68 and 358 bp |
| *CYP2E1* | - | F: 5'- ATT CCC AAG TCC TTC ACC AG -3'<br>R: 5'- GTT GTT TTT GTG CAC CTG GA -3' | qRT-PCR | 60 | 180 bp | - | - | - |
| *GAPDH* | NC_019460.2 | F: 5'-GAG AAA CCT GCC AAG TAT GA-3'<br>R: 5'-TAC CAG GAA ATG AGC TTG AC-3' | qRT-PCR | 60 | 203 bp | - | - | - |
| *ACTB* | NC_019471.2 | F: 5'GAA AAC GAG ATG AGA TTG GC-3'<br>F: 5'CCA TCA TAG AGT GGA GTT CG-3' | qRT-PCR | 60 | 194 bp | - | - | - |

96-well microtiter plate. The reactions were set up using five μL of SYBR® Green Master Mix, 0.5 μL of each forward and reverse primer, two μL of cDNA (50 ng/μl), and two μL of deionized $H_2O$. Each sample was run twice for replication, and the geometric mean of the Ct values was used for mRNA expression profiling. The thermal cycling conditions were as follows: predenaturation at 95˚C for 30 s, 35 cycles of PCR (quantitative analysis) at 95˚C for 5 s and 60˚C for 30 s, melting at 95˚C for 5 s, 60˚C for 1 min, and cooling at 50˚C for 30 s. The target genes were normalized using the housekeeping genes *GAPDH* and *ACTB*. The final results were reported as the delta cycle threshold (ΔCt), calculated as the difference between the target's Ct value and gene housekeeping's Ct value [28]. Differences in *CYP2E1* mRNA expression levels were analyzed with the simple t-test Minitab Software, with confidence interval values of 95% ($P<0.05$) considered to indicate statistically significant differences.

## Immunohistochemistry (IHC) analysis

The (IHC) staining with an anti-CYP2E1 polyclonal antibody (Product PA5-79132, Thermo Fisher Scientific, Waltham, MA, USA) was used to identify the protein expression of CYP2E1. IHC analysis was performed as described by Melzi et al. [29]. The LD was soaked in xylene and alcohol for deparaffinization and rehydration. After washing three times with phosphate buffer saline (PBS) containing 0.05% Tween 20 (PBS-T20), the sections were incubated for 30 min with a blocking buffer containing 10% $H_2O_2$ in methanol to block non-specific binding sites. The primary antibody (dilution 1:200) was added to 50 μL of tissue and incubated overnight at four˚C. Excess primary antibodies were removed by washing with PBS-T20. For primary antibody detection, sections were incubated with the appropriate isotype-specific secondary antibodies conjugated with horseradish peroxidase (HRP) and MACH 1 Universal HRP-Polymer (MRH538L10, Biocare Medical®, Pacheco, CA, USA) for 30 min at 37˚C. This step was followed by a 5 min incubation with 3,3′ -diaminobenzidine (DAB) substrate-chromogen (BDB900G, Biocare Medical®) in 1000 μL Betazoid DAB substrate buffer solution (DS900L10, Biocare Medical®). Tissues were counterstained with hematoxylin and mounted with clear resin and coverslips for long-term storage. We used negative and positive controls to validate the results of IHC staining on muscle tissues. Negative controls were prepared by omitting primary antibodies and replacing them with PBS. The kidney tissues known to express the corresponding antigens were used as a positive control. Tissue sections were screened and photographed under a 40x microscope using the ImageView microphase system, while the anti-CYP2E1 antibody measurements were done using ImageJ software.

## Hematoxylin-eosin (HE) staining

HE staining was carried out to evaluate the morphology of lamb muscle based on the lamb quality value, which, we determined based on the combined pH and tenderness values. Six LDs were selected to represent two quality groups of sheep (three high-quality sheep and three low-quality sheep). HE staining analysis was conducted as described by An et al. [30] and Bao et al. [31]. A $1 \times 2 \times 1$ cm sample was soaked in a 4% paraformaldehyde solution. Muscle tissue preparation was deparaffinized to remove paraffin with xylol, rehydrated with a decreased concentration of alcohol solution, and washed with tap and distilled $H_2O$. The muscle samples were embedded in paraffin wax and sliced into 4-μm thick histological sections. Tissue sections were stained with hematoxylin and eosin. The preparation was observed using a microscope and photographed under 20x and 40x microscope magnification using the ImageView microphase system. Muscle fiber measurement consisted of the diameter and area of the muscle fibers measured from at least 100 μm using ImageJ software.

## Statistical methods

Allele and genotype frequencies were analyzed using genotyping data of seven sheep breeds (see above materials). First, allele and genotype frequencies were calculated using the formula of Nei and Kumar [32]:

$$xi = \frac{(2nii + \sum i \neq j\ nij)}{2N} xii = \frac{nii}{N}$$

Where xi is the frequency of allele i (G and T), xii is the ii frequency of genotype, nii is the number of samples of genotype ij, nij is the number of samples of genotype ij, and N is the total number of samples. Hardy-Weinberg equilibrium [33]:

$$X^2 = \sum \frac{(O - E)^2}{E}$$

X2 is the Chi-Squared, O is the total number of observed genotypes, and E is the total number of expected genotypes.

## Data analysis

The association between the *CYP2E1* gene and carcass and non-carcass characteristics was performed using General Linear Model procedures (Minitab® 18 Software). The model used is as follows:

$$Yij = \mu + Gi + eij$$

Where:
Yijk = the performance of the individual lamb quality traits
μ = the mean of lamb quality traits
Gi = the genotype fixed effect
eij = the random error

## Results

### Genetic polymorphism of *CYP2E1*gene

PCR successfully amplified the DNA fragment of the *CYP2E1* gene. A unique band pattern with an apparent amplicon size of 426 bp represents the amplification product of the *CYP2E1* gene in sheep. After restriction enzyme digestion with *Aci*I, either one band (TT genotype) or a combination of three band patterns indicate homozygous or heterozygous genotypes, respectively. The DNA restriction fragments were obtained for g. 50658168 T>C polymorphism of *CYP2E1-Aci*I was 426 bp for the TT genotype; 426, 358, and 68 bp for the CT genotype; and 68 bp and 358 bp for the CC genotype (Fig 1). Fig 1 shows results that represents only some parts of the amplification PCR-RFLP results from all the samples we observed. Two samples from each breed were chosen to represent the *CYP2E1* gene polymorphism analysis, as shown in Fig 1, and the remaining two samples were JTTS. The CC and CT genotypes were the most frequent (44.5% and 48.5%, respectively) in the investigated flock, whereas the TT genotype was detected in only 14 sheep (7%) (Table 2). The C allele was dominant (68.8%) in the population. All genotypes were found in JTTS, GS, and JS. The TT genotype was not found in JFT, CAS, or BCS. Only the CC genotype was found in BCS, whereas there was no CC genotype in GCS. The chi-square assay revealed that the *CYP2E1* locus was in Hardy-Weinberg equilibrium in our population. The genotype, allele frequency, and chi-square assays of the sheep *CYP2E1* gene are shown in Table 2.

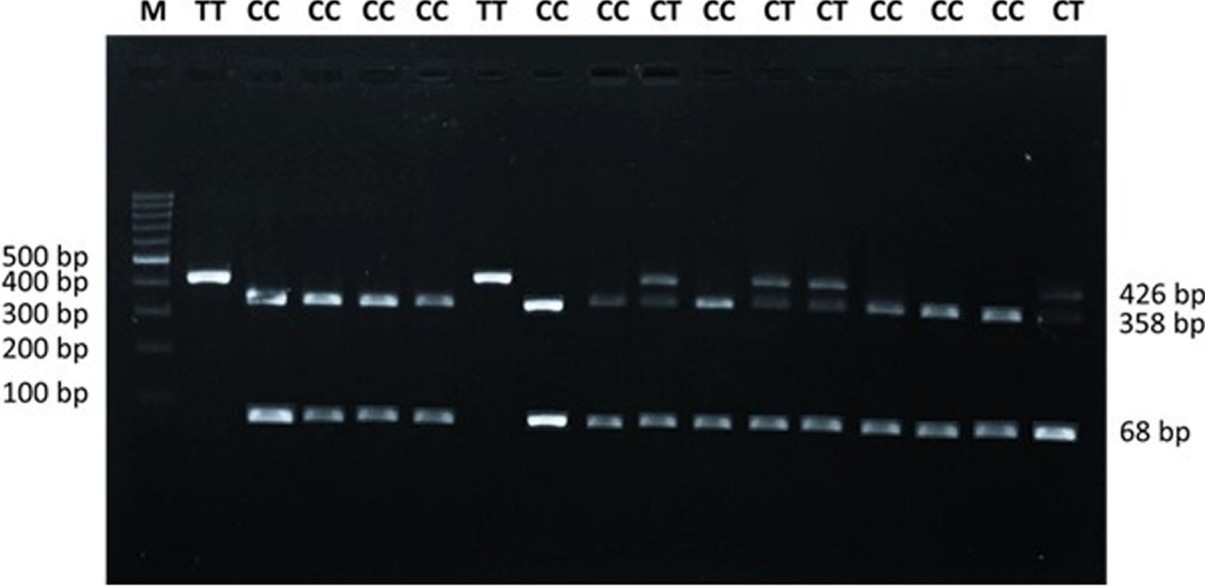

**Fig 1. The visualization *CYP2E1* gene polymorphism.**

## Association study of *CYP2E1* with carcass characteristics, physicochemical lamb, and retail cut

The g. 50658168 T>C of the *CYP2E1* gene had a significant association (P<0.008) with all parameters of carcass traits, i.e., final weight, hot and cold carcass, percentage, and carcass length. In contrast, the *CYP2E1* genotype showed no significant association (P>0.05) with the physicochemical lamb phenotypical traits. The average carcass and physicochemical traits of the CT heterozygotes were significantly higher than those of the CC and TT genotypes for *CYP2E1*. In addition, the final weight ranging from 21.86–27.28 kg, hot carcass (8.48–11.56 kg), cold carcass (8.31–11.26 kg), carcass percentage (39.46–43.62%), and carcass length (8.31–11.26 cm). Furthermore, pH (5.84–6.21), tenderness (35.09–37.58 N), drip loss (45.19–49.29%), and WHC (26.56–28.45%). The carcass characteristics and physicochemical lamb traits are shown in Table 3.

**Table 2. Allele, genotype frequencies, and the chi-square test of the lamb *CYP2E1* gene.**

| Breed | Number of sheep (n) | Genotype Frequency | | | Allele Frequency | | Chi-square test $X^2$ |
|---|---|---|---|---|---|---|---|
| | | CC (n) | CT (n) | TT (n) | A | C | |
| JFTS | 18 | 0.22 (4) | 0.78 (14) | 0.00 (0) | 0.61 | 0.39 | 7.29 |
| JTTS | 105 | 0.42 (44) | 0.55 (58) | 0.03 (3) | 0.69 | 0.31 | 9.68 |
| GS | 20 | 0.60 (12) | 0.35 (7) | 0.05 (1) | 0.77 | 0.23 | 0.00 |
| JS | 27 | 0.37 (10) | 0.41 (11) | 0.22 (6) | 0.57 | 0.43 | 0.75 |
| GCS | 10 | 0.00 (0) | 0.60 (6) | 0.40 (4) | 0.30 | 0.70 | 1.84 |
| CAS | 10 | 0.90 (9) | 0.10 (1) | 0.00 (0) | 0.95 | 0.05 | 0.03 |
| BCS | 10 | 1.00 (10) | 0.00 (0) | 0.00 (0) | 1.00 | 0.00 | 0.00 |
| **Totals** | **200** | **0.45 (89)** | **0.48 (97)** | **0.07 (14)** | **0.69** | **0.31** | **3.31** |

JFTS = Javanese fat-tail sheep; JTTS = Javanese thin-tail sheep; GS = Garut sheep; JS = Jonggol sheep; GCS = Garut composite sheep; CAS = Compass Agrinac sheep; BCS = Barbados cross sheep; N = number of samples, (..) = number of samples which AA, AG, and GG genotype, $\chi^2$ table = 3.84.

**Table 3. The association analyses between *CYP2E1* gene polymorphism and carcass characteristics and chemical lamb meat quality.**

| Traits | *CYP2E1* genotype ($\bar{x}$ ± SE Mean) | | | P-value |
|---|---|---|---|---|
| | **CC**<br>**(n = 33)** | **CT**<br>**(n = 58)** | **TT**<br>**(n = 9)** | |
| **Carcass characteristics** | | | | |
| Final body weight (kg) | 23.21 ± 0.50b | 27.14 ± 0.69a | 24.10 ± 0.69ab | 0.000 |
| Hot carcass (kg) | 9.29 ± 0.28b | 11.39 ± 0.38a | 9.09 ± 0.33b | 0.000 |
| Cold carcass (kg) | 9.265 ± 0.28b | 11.28 ± 0.38a | 9.12 ± 0.35b | 0.000 |
| Carcass percentage (%) | 42.09 ± 0.62ab | 43.87 ± 0.47a | 40.54 ± 0.87b | 0.008 |
| Carcass length (cm) | 63.42 ± 0.85a | 58.77 ± 0.76b | 67.33 ± 1.40a | 0.000 |
| **Retail cut carcass** | | | | |
| **Leg** | 1553.70 ± 43.50ab | 1785.80 ± 48.00a | 1527.40 ± 78.10b | 0.002 |
| Muscle | 1057.20 ± 32.30ab | 1161.00 ± 31.30a | 955.10 ± 66.40b | 0.010 |
| Bone | 430.50 ± 11.50 | 453.89 ± 9.93 | 415.00 ± 30.10 | 0.180 |
| Subcutaneous fat | 55.90 ± 7.70b | 102.40 ± 9.19a | 74.60 ± 5.86ab | 0.002 |
| Intramuscular fat | 23.23 ± 2.69b | 59.52 ± 7.48a | 26.86 ± 1.64ab | 0.001 |
| Pelvic fat | 14.78 ± 1.85b | 23.44 ± 2.28a | 16.90 ± 3.13ab | 0.026 |
| **Loin** | 367.60 ± 16.10b | 464.20 ± 20.90a | 370.20 ± 16.40ab | 0.003 |
| Muscle | 216.15 ± 9.29b | 252.12 ± 9.54a | 180.60 ± 12.70b | 0.002 |
| Bone | 105.54 ± 6.05 | 119.51 ± 6.88 | 119.60 ±9.34 | 0.357 |
| Subcutaneous fat | 28.99 ± 3.98b | 52.58 ± 5.33a | 42.90 ± 12.20ab | 0.011 |
| Intramuscular fat | 11.10 ± 2.35b | 27.33 ± 4.23a | 11.89 ± 4.46ab | 0.012 |
| Pelvic fat | 33.22 ± 3.73ab | 72.27 ± 9.17a | 33.01 ± 5.15b | 0.003 |
| **Flank** | 145.59 ± 6.42b | 194.10 ± 11.10a | 123.40 ± 13.80b | 0.001 |
| Muscle | 106.70 ± 3.73b | 142.82 ± 8.4a | 87.82 ± 8.46b | 0.001 |
| Bone | 0.00 ± 0.00 | 0.00 ± 0.00 | 0.00 ± 0.00 | Nd |
| Subcutaneous fat | 29.72 ± 3.66 | 30.28 ± 2.77 | 29.40 ± 4.01 | 0.988 |
| Intramuscular fat | 4.17 ± 0.74b | 22.37 ± 4.24a | 12.52 ± 4.83ab | 0.005 |
| Pelvic fat | 0.00 ± 0.00 | 0.00 ± 0.00 | 0.00 ± 0.00 | Nd |
| **Shoulder** | 746.10 ± 28.50b | 999.20 ± 39.50a | 818.90 ± 44.40ab | 0.000 |
| Muscle | 505.50 ± 20.90b | 619.00 ± 22.80a | 505.90 ± 25.70ab | 0.002 |
| Bone | 192.4 ± 11.50b | 234.90 ± 10.30a | 203.40 ± 25.00ab | 0.030 |
| Subcutaneous fat | 29.35 ± 2.90b | 50.90 ± 5.51a | 44.60 ± 10.90ab | 0.021 |
| Intramuscular fat | 43.31 ± 5.74b | 83.13 ± 8.15a | 39.71 ± 9.16ab | 0.001 |
| Pelvic fat | 0.00 ± 0.00 | 0.00 ± 0.00 | 0.00 ± 0.00 | Nd |
| **Rack** | 389.60 ± 23.20b | 478.60 ± 20.20a | 401.20 ± 15.10ab | 0.012 |
| Muscle | 231.30 ± 10.00 | 257.80 ± 10.40 | 203.50 ± 18.90 | 0.050 |
| Bone | 134.88 ± 6.21 | 136.19 ± 5.34 | 149.00 ± 8.26 | 0.600 |
| Subcutaneous fat | 27.81 ± 4.47ab | 47.14 ± 5.02a | 22.20 ± 4.01b | 0.010 |
| Intramuscular fat | 10.02 ± 1.48b | 27.76 ± 3.37a | 12.13 ± 3.05ab | 0.000 |
| Pelvic fat | 0.00 ± 0.00 | 0.00 ± 0.00 | 0.00 ± 0.00 | Nd |
| **Breast** | 417.60 ± 14.80b | 552.10 ± 23.30a | 462.90 ± 25.90ab | 0.000 |
| Muscle | 216.30 ± 8.49b | 272.00 ± 11.20a | 223.10 ± 11.10ab | 0.001 |
| Bone | 127.11 ± 4.26b | 147.83 ± 4.87a | 158.39 ± 9.84a | 0.006 |
| Subcutaneous fat | 33.18 ± 3.73b | 56.16 ± 5.41a | 47.93 ± 8.34ab | 0.012 |
| Intramuscular fat | 31.97 ± 2.43b | 64.04 ± 5.96a | 22.51 ± 3.29b | 0.005 |
| Pelvic fat | 0.00 ± 0.00 | 0.00 ± 0.00 | 0.00 ± 0.00 | Nd |
| **Shank** | 388.10 ± 18.00 | 438.30 ± 13.30 | 405.10 ± 22.50 | 0.068 |
| Muscle | 250.30 ± 10.20 | 269.36 ± 9.18 | 246.40 ± 15.00 | 0.315 |

*(Continued)*

**Table 3.** (Continued)

| Traits | CYP2E1 genotype ($\bar{x}$ ± SE Mean) | | | P-value |
|---|---|---|---|---|
| | **CC**<br>**(n = 33)** | **CT**<br>**(n = 58)** | **TT**<br>**(n = 9)** | |
| Bone | 115.36 ± 4.25 | 128.14 ± 3.68 | 122.32 ± 5.48 | 0.084 |
| Subcutaneous fat | 15.44 ± 1.83 | 21.24 ± 1.69 | 22.07 ± 2.59 | 0.065 |
| Intramuscular fat | 9.24 ± 1.05 | 11.17 ± 1.01 | 6.48 ± 2.11 | 0.132 |
| Pelvic fat | 0.00 ± 0.00 | 0.00 ± 0.00 | 0.00 ± 0.00 | Nd |
| **Neck** | 441.50 ± 21.40ab | 552.10 ± 23.10a | 434.10 ± 34.00b | 0.003 |
| Muscle | 281.60 ± 13.80b | 321.00 ± 12.40a | 246.30 ± 23.50b | 0.021 |
| Bone | 131.01 ± 9.34ab | 162.78 ± 8.75a | 130.90 ± 17.90b | 0.043 |
| Subcutaneous fat | 27.69 ± 7.08 | 28.31 ± 2.77 | 21.82 ± 4.51 | 0.819 |
| Intramuscular fat | 16.51 ± 4.03ab | 27.13 ± 3.01a | 6.28 ± 2.63b | 0.010 |
| Pelvic fat | 0.00 ± 0.00 | 0.00 ± 0.00 | 0.00 ± 0.00 | Nd |
| **Totals** | 4546.00 ± 137.00b | 5490.00 ± 185.00a | 4446.00 ± 183.00b | 0.001 |
| **Physicochemical lamb** | | | | |
| pH value | 5.73 ± 0.05 | 5.65 ± 0.02 | 5.74 ± 0.04 | 0.174 |
| Tenderness (N) | 36.55 ± 0.18 | 36.84 ± 0.07 | 38.51 ± 0.12 | 0.673 |
| Cooking loss | 44.75 ± 1.30 | 46.29 ± 1.17 | 50.01 ± 1.05 | 0.224 |
| Water holding capacity | 28.34 ± 0.55 | 28.885 ± 0.42 | 26.51 ± 0.59 | 0.100 |
| **Fatty Acid Composition** | | | | |
| **Fat Content** | 50.40 ± 0.59b | 75.28 ± 0.50a | 21.26 ± 0.21b | 0.000 |
| **Saturated Fatty Acid** | 410.20 ± 1.27 | 419.30± 1.12 | 454.50 ± 1.14 | 0.328 |
| Caprylic acid (C8:0) | 2.08 ± 0.09 | 1.13 ± 0.04 | 1.91 ± 0.08 | 0.494 |
| Capric acid (C10:0) | 10.24 ±10.01 | 6.00 ± 0.31 | 1.24 ± 0.01 | 0.405 |
| Lauric acid (C12:0) | 2.17 ±10.03b | 2.30 ± 0.03b | 6.59 ± 0.16a | 0.000 |
| Tridecylic acid (C13:0) | 0.20 ±10.00 | 1.11 ± 0.08 | 0.20 ± 0.03 | 0.596 |
| Myristic acid (C14:0) | 23.82 ± 0.20 | 24.04 ± 0.14 | 33.17 ± 0.37 | 0.059 |
| Pentadecylic acid (C15:0) | 5.85 ± 0.03 | 5.22 ± 0.026 | 6.12 ± 0.05 | 0.209 |
| Palmitic acid (C16:0) | 183.07 ± 0.92 | 192.98 ± 0.63 | 212.68 ± 0.66 | 0.251 |
| Margaric acid (C17:0) | 6.66 ± 0.07b | 9.12 ± 0.05a | 7.88 ± 0.03ab | 0.013 |
| Stearic acid (C18:0) | 171.63 ± 0.71 | 153.91 ± 0.57 | 182.11± 0.61 | 0.050 |
| Arachidic acid (C20:0) | 10.23 ± 0.27b | 20.04 ± 0.23a | 14.20 ± 0.02b | 0.001 |
| Heneicosylic acid (C21:0) | 2.76 ± 0.06 | 2.00 ± 0.04 | 0.51 ± 0.01 | 0.102 |
| Behenic acid (C22:0) | 0.33 ± 0.01 | 1.10 ± 0.08 | 0.42 ± 0.00 | 0.723 |
| Tricosylic acid (C23:0) | 0.26 ± 0.00 | 0.16 ± 0.00 | 0.15 ± 0.01 | 0.215 |
| Lignoceric acid (C24:0) | 0.10 ± 0.00 | 0.10 ± 0.00 | 0.00 ± 0.00 | 0.374 |
| **Unsaturated Fatty Acid** | 367.20 ± 1.73 | 375.37 ± 0.85 | 368.07 ± 0.71 | 0.874 |
| **Monounsaturated Fatty Acid** | 327.60 ± 1.77 | 349.66 ± 0.91 | 325.86 ± 0.59 | 0.380 |
| Myristoleic acid (C14:1) | 1.68 ± 0.05 | 1.17 ± 0.01 | 1.22 ± 0.02 | 0.446 |
| Palmitoleic acid (C16:1) | 11.12 ± 0.12b | 15.48 ± 0.08ab | 16.50 ± 0.04a | 0.004 |
| Ginkgoleic acid (C17:1) | 6.92 ± 0.51 | 11.59 ± 0.45 | 0.12 ± 0.00 | 0.523 |
| Oleic acid (C18:1n9c) | 271.50 ±1.65 | 284.00 ± 1.07 | 297.20 ± 0.63 | 0.654 |
| Elaidic acid (C18:1n9t) | 32.95 ± 0.17 | 35.62 ± 0.65 | 10.55 ± 0.54 | 0.199 |
| Paullinic acid (C20:1) | 2.58 ± 0.08 | 1.10 ± 0.04 | 0.00 ± 0.00 | 0.057 |
| Erucic acid (C22:1n9) | 0.61 ± 0.02 | 0.57 ± 0.03 | 0.10 ± 0.00 | 0.672 |
| **Polyunsaturated Fatty Acid** | 39.63 ± 0.27a | 25.71 ± 0.22b | 42.21 ± 0.35a | 0.000 |
| Linoleic acid (C18:2n6c) | 20.01 ± 0.23a | 10.07 ± 0.16b | 25.44 ± 0.20a | 0.000 |
| y-linolenic acid (C18:3n6) | 0.66 ± 0.01 | 0.87 ± 0.01 | 0.26 ± 0.02 | 0.073 |

*(Continued)*

**Table 3.** (Continued)

| Traits | CYP2E1 genotype ($\bar{x}$ ± SE Mean) | | | P-value |
|---|---|---|---|---|
| | CC (n = 33) | CT (n = 58) | TT (n = 9) | |
| α-linolenic acid (C18:3n3) | 4.03 ± 0.06b | 3.87 ± 0.03b | 6.93 ± 0.07a | 0.017 |
| Eicosedienoic acid (C20:2) | 0.49 ±0.01 | 0.82 ± 0.05 | 0.37 ± 0.00 | 0.790 |
| Dihomo-y-linolenic acid (C20:3n6) | 0.06 ±0.01 | 0.72 ± 0.04 | 0.48 ± 0.00 | 0.936 |
| Arachidonic acid (C20:4n6) | 5.35 ± 0.06 | 3.90 ± 0.05 | 3.71 ± 0.03 | 0.330 |
| Docosadienoic acid (C22:2) | 0.60 ± 0.00 | 0.10 ± 0.00 | 0.00 ± 0.00 | 0.304 |
| Eicosapentaenoic acid (C20:5n3) | 5.31 ±0.07a | 2.33 ± 0.03b | 3.84 ± 0.05ab | 0.000 |
| Cervonic acid (C22:6n3) | 0.62 ±0.01 | 0.42 ± 0.01 | 0.51 ± 0.01 | 0.362 |
| **Fatty Acid Total** | 784.60 ± 22.00 | 806.27 ± 8.60 | 822.60 ± 16.70 | 0.404 |
| **Cholesterol** | 109.10 ± 1.63a | 63.60 ± 0.26b | 82.50 ± 0.55ab | 0.002 |
| **Flavor Odor** | | | | |
| 4-methyl octanoic acid (MOA) | 430.00 ± 27.40 | 72.50 ± 37.80 | 112.60 ± 50.40 | 0.210 |
| 4-ethyloctanoic acid (EOA) | 108.90 ± 39.00 | 146.70 ± 52.90 | 106.60 ± 35.30 | 0.854 |
| 4-methyl nonanoic acid (MNA) | 1220.00 ± 56.20 | 560.00 ± 27.90 | 2111 ± 12.55 | 0.207 |
| 3-methylindole (MI) | 0.08 ± 0.03b | 1.71 ± 0.58ab | 5.64 ± 3.64a | 0.007 |
| 3-methyl phenol (MP) | 129.00 ± 11.50 | 22.30 ± 11.70 | 1.00 ± 0.73 | 0.412 |
| 4-methyl phenol (MP) | 1.086 ± 0.45 | 1.74 ± 0.70 | 1.44 ± 0.75 | 0.793 |
| **Mineral content** | | | | |
| Iron (Fe) | 1.75 ± 0.11 | 1.93 ± 0.11 | 1.86 ± 0.28 | 0.584 |
| Zinc (Zn) | 2.63 ± 0.21 | 2.59 ± 0.10 | 2.65 ± 0.30 | 0.978 |
| Potassium (K) | 290.50 ± 12.70 | 263.70 ± 11.00 | 280.30 ± 44.40 | 0.353 |
| Selenium (Se) | 0.53 ± 0.05 | 0.65 ± 0.04 | 0.73 ± 0.04 | 0.095 |

Note: $\bar{x}$ = the average of traits; SE = standard error; Superscript indicated a significant difference at 5% (P<0.05); Nd = not detected; Numbers shown in parentheses are the number of individuals with the specified genotype.

In addition, the association analysis of *CYP2E1* gene polymorphisms with retail cut carcasses revealed a significant association (P<0.05) with leg, shoulder, rack, breast, loin, flank, and neck (Table 3). The CT genotype had a higher carcass percentage and retail cut carcass composition than the CC and TT genotypes. The carcass fat from CT heterozygotes was also higher than the others. The fat in carcasses consists of subcutaneous, intramuscular, and pelvic fat. All carcass cut parameters were subcutaneous and intramuscular fat. However, pelvic fat is only found in leg and loin cuts.

## Association study of *CYP2E1* with fatty acid composition

The *CYP2E1* gene polymorphism (g. 50658168 T>C) was significantly associated (P<0.05) with fatty acids, including fat content, SFA, particularly lauric acid (C12:0), margaric acid (C17:0), stearic acid (C18:0), and arachidic acid (C20:0). Furthermore, *CYP2E1* also had a significant association with MUFA consisting of palmitoleic acid (C16:1) and PUFA composed of linoleic acid (C18:2n6c), α-linolenic acid (C18:3n3), and eicosapentaenoic acid (C20:5n3). Generally, the total UFA content was marginally higher in the CT genotype (375 mg/100 g) than in the CC (367 mg/100 g) and TT (368 mg/100 g) genotypes. The fatty acid content of lambs is presented in Table 3.

## Association study of *CYP2E1* with the cholesterol content

The *CYP2E1* gene polymorphism g. 50658168 T>C significantly affected the cholesterol content (P<0.05). Sheep with the CT heterozygote genotype had a significantly lower cholesterol content (63 mg/100 g) than those with the CC and TT genotypes (109 and 82 mg/100 g, respectively). Table 3 presents the association between the *CYP2E1* gene and cholesterol content.

## Association study of *CYP2E1* with flavor and odor

Association analysis of *CYP2E1* (g. 50658168 T>C) gene polymorphism with flavor and odor content revealed a significant association (P<0.05) with 3-methyl indole (3-MI) (Table 3). CC homozygote lambs had a lower MI (0.08 μg/μl) when compared to either CT heterozygotes (1.71 μg/μl) or CT homozygotes (5.64 μg/μl). In addition, other flavor and odor traits such as 4-methyl octanoic acid (MOA) (72.5–430 μg/μl), 4-methyl octanoic acid (EOA) (106.60–146.70 μg/μl), 4-methyl nonanoic acid (MNA) (560–2111 μg/μl), 3-methylindole (MI) (0.08–5.64 μg/μl), 3-methyl phenol (3-MP) (1.00–129.00 μg/μl), and 4-methyl phenol (4-MP) (1.44–1.74 μg/μl), were not shown to have a strong association to the *CYP2E1* gene polymorphism. The association analysis between *CYP2E1* gene polymorphism and flavor and odor is shown in Table 3.

## Association study of *CYP2E1* with mineral content

*CYP2E1* polymorphism results (g. 50658168 T>C) analysis and their association with mineral content are presented in Table 3. *CYP2E1* was not associated with mineral content (P>0.05). Moreover, minerals consisting of Fe (1.75–1.93 mg/ 100 g), Zn (2.59–2.65 mg/100 g), K (263.70–280.30 mg/100 g), and Se (0.52–0.73 mg/100 g).

## The mRNA expression level of the *CYP2E1* gene by qRT-PCR

The mRNA expression levels of *CYP2E1* were significantly different (P<0.05) between genotypes. The CT genotype had a higher expression level than the CC and TT genotypes (Fig 2). The mRNA expression results of qRT-PCR appeared to be consistent with the association study results, which showed that the sheep carrying the CT genotype had a higher characteristic and retail cut carcass, physicochemical lamb, UFA, and mineral content, and lower SFA, cholesterol value, flavor, and odor compounds. These results indicated that sheep carrying the CT genotype would produce a healthy lamb.

## Localization of CYP2E1 protein in LD tissues by IHC staining

IHC staining analysis showed that the CYP2E1 protein in the LD muscle cells was detected in the cytoplasm (Fig 3). The expression of CYP2E1 in the LD was significantly lower than in other cellular tissues, such as the kidney.

## Comparison of morphological differences of LD based on physicochemical traits by HE staining

The morphology of the LD by HE staining showed that the muscle fibers were long and fusiform, causing the transverse cut section (Fig 4). The difference between the two groups in muscle fiber area and diameter showed that the group with high pH and shear force had more muscle fiber area and diameter than the low group, resulting in the muscle cells in the high group absorbing more red color (hema) from the HE staining.

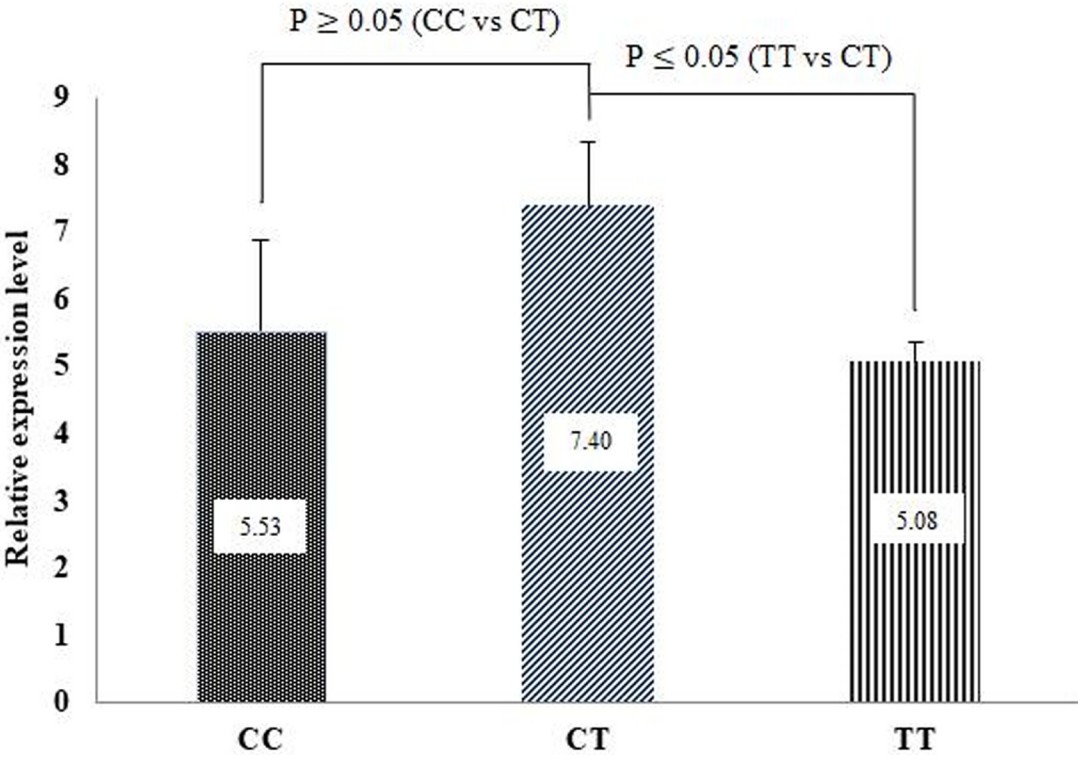

**Fig 2. The expression levels of *CYP2E1* in lamb quality.**

## Discussion

### *CYP2E1* gene polymorphism and its association with lamb meat quality

Lamb meat contains nutrients essential for human health, such as fatty acids [34,35], flavor odors [36], cholesterol [37], and minerals [34,38]. Therefore, nutritional quality is a crucial marketing tool to compete with meat from other animals. Lamb meat quality is a polygenic quantitative trait [39]. The CYP2E1 protein plays a vital role in meat quality. Previously, we found that genetic variation in the *CYP2E1* gene was associated with lamb quality at SNP g.50657948 T>G [16]. In the current study, we identified a new SNP, g. 50658168 T>C in the *CYP2E1* gene and its association with lamb meat quality traits, including carcass characteristics, retail cut carcasses, physicochemical properties of lamb, fatty acid, cholesterol, flavor and

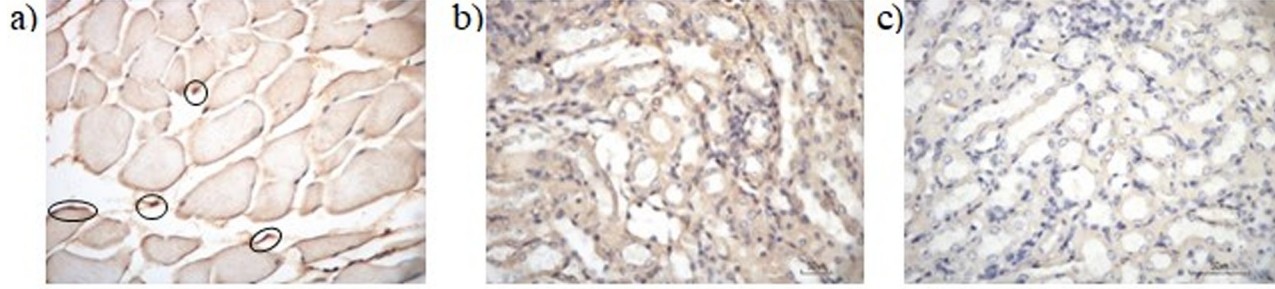

**Fig 3. The CYP2E1 protein expression.** (a) LD muscle; (b) control positive using anti-CYP2E1 antibodies in the different sample (kidney); (c) negative control without anti-CYP2E1 antibodies in kidney sample.

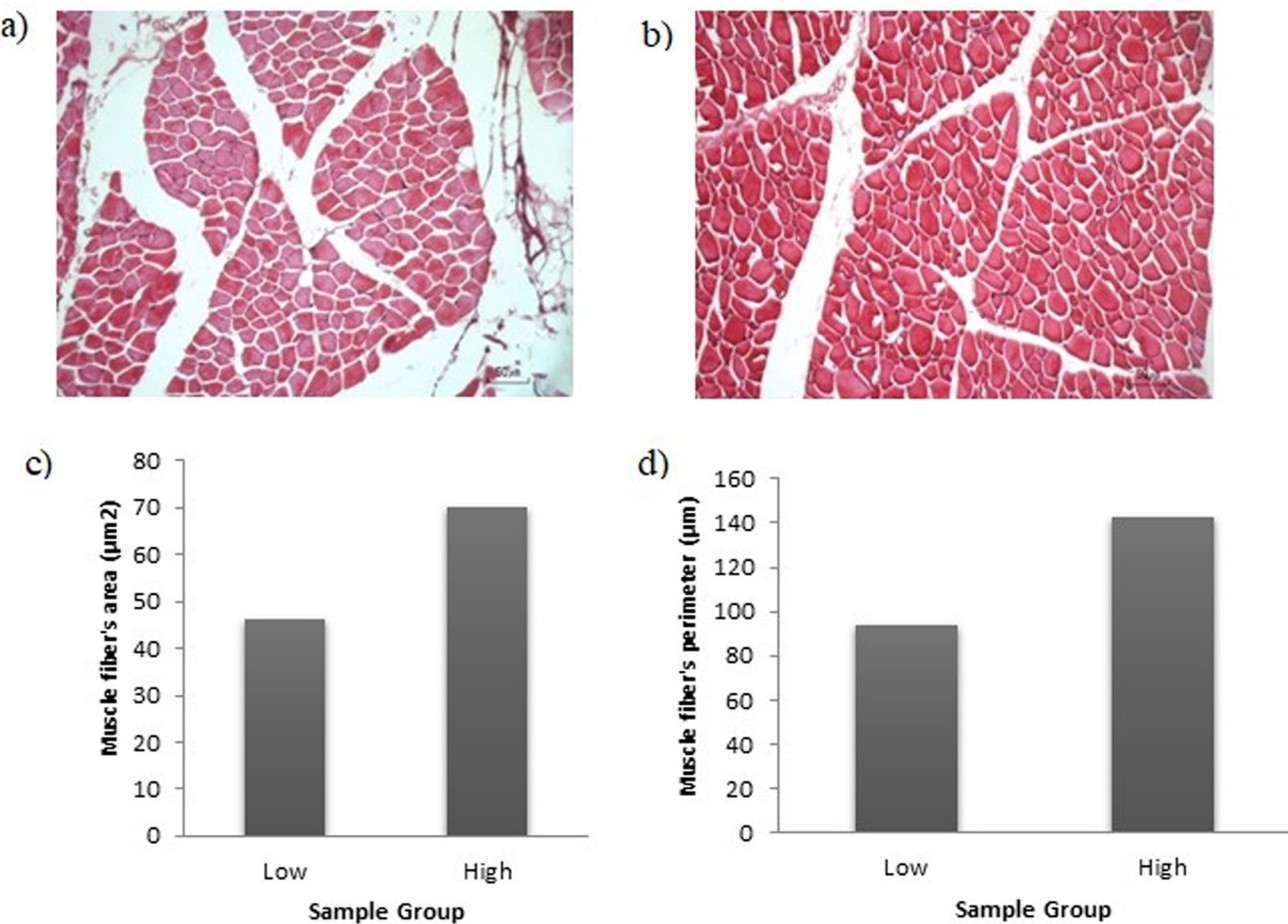

**Fig 4. Morphological characteristics of HE-stained.** (A) The morphology of the low group of lamb quality; (B) The morphology of the high group of lamb quality; (C) Muscle fiber's area in different groups; (D) Muscle fiber's perimeter in different groups.

odor, and mineral content, as well as the protein expression of CYP2E1 in muscle by IHC and HE staining. The *CYP2E1* PCR product used to detect the genotypes for the new SNP, g. 50658168 T>C was detected in gDNA obtained from all sheep samples used in our study. The combination of bands revealed by PCR-RFLP for the *CYP2E1* genotype is one band at 426 bp for the TT genotype, two bands at 368 and 68 bp for the CC genotype, and three bands at 426, 368, and 68 bp for the CT genotype. The genetic variant analyzed in this study was a T-to-C transition. The SNP g. 50658168 T>C of *CYP2E1* genetic variant is located in the 3'-UTR and could potentially influence miRNA targeting efficiency. Accordingly, combining the T and C alleles forms three genotypes: homozygous TT, homozygous CC, and heterozygous CT. Three *CYP2E1* genotypes were found in JTTS, GS, and JS. Heterozygous CT was dominant and found in all sheep breeds in our study except for the BCS breed. All sheep breeds with BCS had homozygous CC. In addition, homozygous TT was considered rare and was only found in 7% of JTTS, GS, JS, and GCS. GCS, CAS, and BCS crossbred sheep were developed by the Center for Research and Development of Animal Husbandry, Indonesia, which had a selection. Selection and crossing can cause uneven genotype distribution of a gene. Harahap et al. [40] reported that gene polymorphism was found evenly in BCS, CAS, and GCS, and even Abdillah et al. [41] reported that the *CYP2A6* gene was monomorphic in those breeds. Generally, the

SNP g. 50658168 T>C *CYP2E1* was in Hardy-Weinberg equilibrium in our sheep study population.

The sheep in this study exhibited notably high body weights, ranging from 23.21–27.14 kg, and produced hot carcasses from 9.09–11.39 kg. Therefore, the percentage of carcasses produced ranged between 40.54–43.87%. These results were higher than those reported by Saeed et al. [42], indicating that the carcass's dressing percentage is 38.00–39.69%. The portion of the carcass in this study was higher than that observed by Saeed et al. [42] but based on body weight at the same-aged slaughter, the sheep had higher body weight than in this study. The carcass weight of the lambs significantly affected their retail cut composition. High carcass weight produces a high retail cut. The primary cuts of carcass lamb consisted of the leg (32.94%), shoulder (25.22%), rack (12.41%), and loin (11.05%), while the lower value cuts were the neck (5.30%) and breast (3.55%) of sheep with a body weight of 25.32 kg [43]. In this study, subcutaneous fat, one of the main attributes used to evaluate meat quality, was higher than intramuscular fat (IMF). However, IMF was correlated with palatability, juiciness, tenderness, and higher flavor precursors with an increase in fat content [44], and lower IMF indicated an adverse effect on eating satisfaction [45]. Morever, the non-carcass components of sheep can cause differences in the percentage of carcasses produced. The high non-carcass part resulted in a lower carcass component. The percentage of total non-carcass components as a slaughtering product is between 44.8%–46.4% [46], while carcass components range from 36% to approximately 60%.

However, the *CYP2E1* polymorphism showed no significant effect on physicochemical lamb. The pH value was in the normal pH (5.4–5.7). Knight et al. [35] reported that the pH ultimate of lamb is 5.7. The postmortem phase affected the pH value during the slaughtering process. Ge et al. [47] divide pH value into pre-rigor, rigor-mortis, and post-rigor, with values of 6.50–6.54, 5.63–5.92, and 5.63–5.96, respectively. The pH is related to lamb tenderness, water-holding capacity, and cooking loss. The rate of pH decline is vital as it dictates meat tenderness. Cold shortening can occur if the pH is high, muscle temperatures decrease rapidly, and the meat becomes tough [48]. Low pH causes water loss from the meat (i.e., decreased water-holding capacity) and a lighter color. A high pH value results in a darker color and less drip loss, often associated with poor lamb eating quality [49].

The *CYP* gene family has been reported in many studies on fatty acids [16,27]. The meat with the CT genotype of *CYP2E1* had a higher UFA than the TT genotype. The total SFA in this study was higher than that reported by Bravo-Lamas et al. [44], who reported SFA, MUFA, and PUFA values of 394 mg/g, 379 mg/g, and 137 mg/g, respectively. In this study, total SFA, MUFA, and PUFA were 410–454 mg/g, 325–349 mg/g, and 25.7–42.2 mg/g, respectively. Oleic, palmitic, and stearic acids dominated the levels of FA. Notably, the FA profile affected the sensory quality of the lambs. Most volatile carbonyl compounds are generated from the oxidation of UFA and can be potent odorant compounds [50]. However, fatty acids, primarily obtained from red meat, are essential nutrients for the human diet. These fatty acids' health benefits have been reported to decrease cancer and cardiovascular risks [51].

Generally, fatty acids are correlated with cholesterol. Lamb meat has a lower cholesterol content than beef meat. The cholesterol content in our study was higher than that reported by Wati et al. [52], which was 26.24–37.67 mg/100 g. Cadavez et al. [53] also noted that the cholesterol content of lamb meat was 40.97 and 66.88 mg/100 g of meat, respectively. Another cholesterol content value reported by Coutinho et al. was 71.22–78.31 mg/100 g [54]. The cholesterol content in the longissimus sample was higher values of 152, 590, and 809 mg/100 g [45].

Furthermore, we found that the *CYP2E1* gene polymorphism was significantly associated with flavor and odor, namely 3-methyl indole (3-MI). The lambs of the CC homozygotes had

lower 3-MI levels than lambs with the CT or TT genotypes.Similarly, Harahap et al. [16] reported that the SNP g.50657948 T>G of *CYP2E1* was associated with MI in lamb quality. The range of MI is between 0.128 and 0.870 ug/g. These low flavor and odor intensity scores may be attributed to the low intramuscular fat content of the meat. The CYP2E1 protein is involved in the metabolism of 3-methylindole, one of the significant contributors to "boar taint" in meat from uncastrated male pigs [55]. 3-methyl indole (3-MI), also called skatole, is one of the branched-chain fatty acids, besides 4-methyl phenol, related to the 'pastoral' flavor. In contrast, 4-methyl octanoic (MOA), 4-methyl octanoic (EOA), and 4-methyl nonanoic acids were compounds related to 'mutton flavor' in cooked sheep meat [56]. Insausti et al. [57] reported that aging lambs might positively affect meat's sensory odor and flavor quality. However, the *CYP2E1* was found not to correlate to mineral content.

## The *CYP2E1* mRNA expression

*CYP2E1* gene is a member of the oxide reductase cytochrome gene family that encodes enzymes that can oxidize a variety of small-molecule endogenous substrates, such as biosynthesis and conversion cholesterol to steroid hormones, androgens, estrogens, and omega hydroxylation of fatty acids [17,18]. The role of CYP2E1 protein in oxidation is essential for its effect on meat quality. CYP2E1 has been widely reported to regulate androstenone and skatole in porcine. Skatole was produced by bacterial fermentation of the amino acid l-tryptophan in the large intestine and is metabolized by liver enzymes responsible for sulfation and oxidation [58]. CYP2E1 activity in liver tissues significantly affects skatole concentrations and androsterone levels in the fat [59]. Other specific enzymes, CYP2A, were identified as the main enzymes involved in the metabolism of skatole phase 1 [60].

Our study's *CYP2E1* mRNA expression levels showed significantly different transcript abundances in the liver between genotypes. The CT genotype of *CYP2E1* had the highest expression level compared to the CC and TT genotypes, which could be due to breed differences in the analyses. Similarly, Kubesova et al. [61] reported that the mRNA levels of *CYP2E1* in pigs significantly differed. In our study, the *CYP2E1* mRNA was upregulated in lamb, showing favorable meat carcass characteristics and physical properties. These results are consistent with an association study of the *CYP2E1*, which showed that the CT genotype had a higher characteristic and retail cut carcass, physicochemical lamb, UFA, and mineral content and lower SFA, cholesterol value, flavor, and odor compounds. These results indicated that sheep inheriting the CT genotype would produce healthier lamb meat. However, the *CYP2E1* expression level in our study might be biased due to the limited number of samples used. The results of our study need to be further confirmed in a more extensive and diverse population.

## Localization of CYP2E1 protein using IHC and HE staining

Herein, we first examined CYP2E1 expression at the protein level to determine its association with lamb quality traits. The CYP2E1 protein was found to be expressed in LD muscle cells by IHC analysis. CYP2E1 is located in the cytoplasm, especially the reticulum endoplasm. We used other cellular tissues like the kidney to control protein expression. Moreover, CYP2E1 protein expression was lower in the LD than in the control group. CYP2E1 protein expression and related activities are often regulated at the posttranscriptional level. Rasmussen and Zamaratskaia [62] reported that changes in porcine *CYP2E1* mRNA expression are not always reflected in the corresponding *CYP2E1* protein expression and activity changes. Therefore, *CYP2E1* mRNA and CYP2E1 protein expression in this study were confirmed to be associated with lamb quality in Indonesian sheep. CYPs can be divided into different families according to their amino acid composition. CYP2 family members, including CYP2E1, played an

essential role in drug metabolism, in addition to CYP1 and CYP3 families. CYP2E1 protein expression in pigs is higher in fetuses with low birth weight in the porcine liver [63].

Microanatomical observations of the LD muscle stained with HE showed differences in the transverse sectional area and perimeter of the muscle in lamb meat. Muscle samples with high pH and WBSF values had higher area and perimeter values than those with low pH values. Furthermore, the muscle cells in the high group absorbed more red color (hema), as determined by HE staining. A high WBSF value indicates that the meat will be less tender. Kiran et al. [64] and Nuraini et al. [65] stated that the cross-sectional area of the muscle and the higher the amount of muscle can result in lower tenderness, higher water-binding capacity, and lower cooking losses because the liquid contained in power is less likely to come out of the meat. Morphologically, skeletal muscle tissues include the epimysium, endomysium, myofibrils, blood vessels, nerves, muscle fibers, and muscle fascicles [66]. The fascicle size is related to the cross-sectional area of the muscle, the number of muscles per bundle, the amount of connective tissue, and the distance between the muscles. Connective tissue affects tenderness through its structure and composition [66]. In particular, collagen is generally considered the primary determinant of shear force. The interaction between muscle fibers and collagen modulates the thermal denaturation of collagen. The structure of connective tissue, particularly its organization and size of the perimysium bundles, plays a role in developing meat texture [67].

## Conclusion

The genotype frequencies of three genotypes, TT, CC, and CT, at the newly discovered SNP locus g. 50658168 T>C of the *CYP2E1* gene ranged between 0.07–0.48 in the seven Indonesian sheep breeds analyzed here. The *CYP2E1* gene polymorphisms were significantly associated (P<0.05) with lamb quality traits, including carcass characteristics, retail cut carcass, fatty acid, cholesterol, flavor, and odor. The CT genotype had a higher expression level of lamb quality traits than the CC and TT genotypes of Garut composite sheep. The CT genotype had a higher characteristic: retail cut carcass, physicochemical lamb, UFA, mineral content, lower SFA, cholesterol value, flavor, and odor compounds. CYP2E1 protein expression is localized to the LD muscle. The group with a high pH had a higher area and perimeter of the muscle cells. Sheep that inherit the CT genotype produces a healthy lamb. Therefore, *CYP2E1* can be used as a genetic marker for selecting sheep with high meat quality. However, further validation is still needed to confirm the effect of *CYP2E1* gene expression and polymorphisms in different sheep populations.

## Supporting information

**S1 Table. The genotype of the *CYP2E1* gene in Indonesian sheep.**
(XLSX)

**S2 Table. Carcass characteristics and chemical lamb meat quality.**
(XLSX)

**S3 Table. The qRT-PCR results.**
(XLSX)

**S4 Table. The data of muscle fiber's area and perimeter for HE staining analysis.**
(XLSX)

## Author Contributions

**Conceptualization:** Ratna Sholatia Harahap, Asep Gunawan.

**Data curation:** Ratna Sholatia Harahap.

**Formal analysis:** Ratna Sholatia Harahap, Yuni Cahya Endrawati, Huda Shalahudin Darusman, Ronny Rachman Noor.

**Funding acquisition:** Asep Gunawan, Ronny Rachman Noor.

**Investigation:** Ratna Sholatia Harahap.

**Methodology:** Ratna Sholatia Harahap, Huda Shalahudin Darusman.

**Project administration:** Asep Gunawan, Ronny Rachman Noor.

**Resources:** Asep Gunawan.

**Software:** Ratna Sholatia Harahap.

**Supervision:** Yuni Cahya Endrawati, Huda Shalahudin Darusman, Göran Andersson, Ronny Rachman Noor.

**Validation:** Yuni Cahya Endrawati.

**Visualization:** Ratna Sholatia Harahap, Göran Andersson.

**Writing – original draft:** Ratna Sholatia Harahap.

**Writing – review & editing:** Asep Gunawan, Göran Andersson, Ronny Rachman Noor.

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
