## [Decision Letter · Decision Letter 0]

8 Apr 2024

PONE-D-24-10259A comprehensive study of CYP2E1 and its role in carcass characteristics and chemical lamb meat quality in different Indonesian sheep breedsPLOS ONE

Dear Dr. Noor,

Thank you for submitting your manuscript to PLOS ONE. After careful consideration, we feel that it has merit but does not fully meet PLOS ONE’s publication criteria as it currently stands. Therefore, we invite you to submit a revised version of the manuscript that addresses the points raised during the review process.

We look forward to receiving your revised manuscript.

Kind regards,

Palash Mandal

Academic Editor

PLOS ONE

Journal Requirements:

“This research was funded by The Directorate General of Resources for Science, Technology, and Higher Education, Ministry of Research, Technology, and Higher Education, grant number 077/SP2H/LT/DRPM/2021, dated March 18, 2021 and Post-Doctoral Project 2024 IPB University with grant number 3/IT3.D12/SP/DAPT.PD/2024, dated February 28, 2024. The funders had no role in the study design, data collection, analysis, publication decisions, or manuscript preparation.”

“-AG and RSH received an award from The Directorate General of Resources for Science, Technology, and Higher Education, Ministry of Research, Technology, and Higher Education, with grant number: 077/SP2H/LT/DRPM/2021, dated March 18, 2021

- RRN and RSH received an award from Post-Doctoral Project 2024 IPB University with grant number: 3/IT3.D12/SP/DAPT.PD/2024, date February 28, 2024.

-The funders had no role in the study design, data collection, analysis, publication decisions, or manuscript preparation.”

Additional Editor Comments (if provided):

Dear Editor,

As I got conflicting reports from the reviewers, then myself reviewed the manuscript. After careful consideration, I feel that it has merit but does not fully meet PLOS ONE publication criteria as it currently stands. The shortcomings of this paper needs to be worked out before it can be considered for publication. Therefore, I would like to invite them to resubmit a revised version of the manuscript that addresses the points raised during the review process.

For your guidance, the reviewers' comments are included below.

Specific concerns expressed during peer review were:

Specific concerns expressed during peer review were:

Comments from Reviewer 1

The manuscript is sound and worthy of publishing, although the research is targeted towards a specific group of researchers. The English needs to be revised. However, it is not clear which genotype is desired and why. Tables 2 to 8 describe similar association data, so they could be merged into a single table. Why was mRNA expression performed in tissues with different genotypes, while protein localization was performed in kidney and muscle tissues? Does the lower protein expression in muscle compared to the kidney hold any significance? What is the relationship between the data in Figure 4 and CYP2E1? Additionally, why were liver tissues used for mRNA expression while phenotypes were measured in muscle tissues?

Comments from Reviewer 2

Major comments:

L129: What is the definition of concentrate GTO3? Kindly specify in detail, avoiding from abbreviations. Furthermore, there is evidence to suggest that the JTTS oxen used in the slaughterhouse were obtained from multiple farms located in West Java, Indonesia, and were fed in group cages. Nonetheless, the descriptions given are extremely general. It is imperative to specify the types of forage and concentrated feed employed, as well as the duration of their administration.

L159-161: Cooked meat samples are typically employed for Warner-Bratzler Shear Force (WBSF) analysis. Kindly elaborate on the preparatory procedures preceding WBSF. In cases where WBSF was conducted on raw meat, as reported in certain studies, please specify this approach.

L175: Please specify the column utilized for the fatty acid analysis and provide relevant properties. A similar situation is valid for cholesterol, flavor and odor measurements.

L198-200: Which instrument did you employ for the mineral content analysis of the LD muscle, the ICP-OES or the AA spectrophotometer? Please provide specific details.

L219-220: In your study, it was noted that JTTS rams were utilized, with a distribution of CC (n=33), CT (n=58), and TT (n=9) genotypes. Could you elaborate on the rationale and methodology employed for obtaining liver tissue samples from Garut composite sheep rams?

L261-262: In determining the grade of sheep groups, what specific criteria are utilized? Please share extra specific details about your methods.

L520-522: An apparent discrepancy is evident in either the materials and methods or in the discussions: While JTTS rams were indicated to have been used for slaughter in the materials and methods, there appears to be a contradiction in attributing the observed variances to breed differences.

Minor comments:

Please verify that abbreviations used throughout the text are appropriately placed.

L25: CYP2E1 appears in numerous places throughout the text, both in italics and in normal form. Each must be italicized for correction.

L26: Please use “g. 50658168” instead of “50658168”.

L28, L39, L148, L429: Please use “fatty acid” instead of “fat acid”.

L60, L61: Please use “grams/capita/year” instead of “grams per capita per year”.

L313, L335: Please ensure that each abbreviation used in the table is accompanied by a corresponding footnote below the table

L340: Please use “fat” instead of “Fats” in Table 4.

L353: Please provide the section where lipid numbers are presented within parentheses.

L386: Please use “potassium (K)” instead of “Kalium (K)”..

Reviewers' comments:

Reviewer's Responses to Questions

**Comments to the Author**

1. Is the manuscript technically sound, and do the data support the conclusions?

Reviewer #1: Yes

Reviewer #2: Partly

2. Has the statistical analysis been performed appropriately and rigorously? 

Reviewer #1: Yes

Reviewer #2: Yes

3. Have the authors made all data underlying the findings in their manuscript fully available?

Reviewer #1: Yes

Reviewer #2: Yes

4. Is the manuscript presented in an intelligible fashion and written in standard English?

Reviewer #1: Yes

Reviewer #2: Yes

5. Review Comments to the Author

Reviewer #1: The manuscript is sound and worthy of publishing, although the research is targeted towards a specific group of researchers. The English needs to be revised. However, it is not clear which genotype is desired and why. Tables 2 to 8 describe similar association data, so they could be merged into a single table. Why was mRNA expression performed in tissues with different genotypes, while protein localization was performed in kidney and muscle tissues? Does the lower protein expression in muscle compared to the kidney hold any significance? What is the relationship between the data in Figure 4 and CYP2E1? Additionally, why were liver tissues used for mRNA expression while phenotypes were measured in muscle tissues?

Reviewer #2: Major comments:

L129: What is the definition of concentrate GTO3? Kindly specify in detail, avoiding from abbreviations. Furthermore, there is evidence to suggest that the JTTS oxen used in the slaughterhouse were obtained from multiple farms located in West Java, Indonesia, and were fed in group cages. Nonetheless, the descriptions given are extremely general. It is imperative to specify the types of forage and concentrated feed employed, as well as the duration of their administration.

L159-161: Cooked meat samples are typically employed for Warner-Bratzler Shear Force (WBSF) analysis. Kindly elaborate on the preparatory procedures preceding WBSF. In cases where WBSF was conducted on raw meat, as reported in certain studies, please specify this approach.

L175: Please specify the column utilized for the fatty acid analysis and provide relevant properties. A similar situation is valid for cholesterol, flavor and odor measurements.

L198-200: Which instrument did you employ for the mineral content analysis of the LD muscle, the ICP-OES or the AA spectrophotometer? Please provide specific details.

L219-220: In your study, it was noted that JTTS rams were utilized, with a distribution of CC (n=33), CT (n=58), and TT (n=9) genotypes. Could you elaborate on the rationale and methodology employed for obtaining liver tissue samples from Garut composite sheep rams?

L261-262: In determining the grade of sheep groups, what specific criteria are utilized? Please share extra specific details about your methods.

L520-522: An apparent discrepancy is evident in either the materials and methods or in the discussions: While JTTS rams were indicated to have been used for slaughter in the materials and methods, there appears to be a contradiction in attributing the observed variances to breed differences.

Minor comments:

Please verify that abbreviations used throughout the text are appropriately placed.

L25: CYP2E1 appears in numerous places throughout the text, both in italics and in normal form. Each must be italicized for correction.

L26: Please use “g. 50658168” instead of “50658168”.

L28, L39, L148, L429: Please use “fatty acid” instead of “fat acid”.

L60, L61: Please use “grams/capita/year” instead of “grams per capita per year”.

L313, L335: Please ensure that each abbreviation used in the table is accompanied by a corresponding footnote below the table

L340: Please use “fat” instead of “Fats” in Table 4.

L353: Please provide the section where lipid numbers are presented within parentheses.

L386: Please use “potassium (K)” instead of “Kalium (K)”.

6. PLOS authors have the option to publish the peer review history of their article (what does this mean?). If published, this will include your full peer review and any attached files.

Reviewer #1: **Yes: **Jasim Muhammad Uddin

Reviewer #2: No

---

## [Author Response · Author response to Decision Letter 0]

7 May 2024

We sincerely appreciate your time and effort in considering and reviewing our manuscript for publication in PLOS ONE. We have revised our manuscript, and below is our response to the reviewers' comments and suggestions.

Comments from Reviewer 1:

Comment 1: The manuscript is sound and worthy of publishing, although the research is targeted towards a specific group of researchers.

Response 1: Thanks for the consideration and review of our manuscript.

Comment 2: The English needs to be revised.

Response 2: We appreciate the reviewer's comments. An English-speaking expert in animal science has checked the manuscript. All changes and corrections are in the revised manuscript with track changes.

Comment 3: However, it is not clear which genotype is desired and why. 

Response 3: Thanks for the comment. Based on an association study analysis between the CYP2E1 gene and lamb quality, the results showed that the CT genotype had the best value for lamb quality traits: retail cut carcass, physicochemical lamb, unsaturated fatty acid, mineral content, lower saturated fatty acid, cholesterol value, flavor, and odor compounds. We have added this point in the abstract and conclusion.

Comment 4: Tables 2 to 8 describe similar association data, so they could be merged into a single table. 

Response 4: Thanks for the suggestions. For the analysis results, we have combined tables 3 to 8 into one comprehensive table, which we make in Table 3. Please see line 396 in the revised manuscript with track changes.

Comment 5: Why was mRNA expression performed in tissues with different genotypes, while protein localization was performed in kidney and muscle tissues?

Response 5: Thanks for the question. We analyzed mRNA expression based on genotype variations to see the CYP2E1 gene expression. Next, to validate this mRNA expression, we analyzed CYP2E1 protein expression directly in muscle samples to see whether CYP2E1 expression was also present in the sheep muscle samples we used. We validated this protein's expression using positive and negative controls. The positive control was used from kidney tissues because the CYP2E1 protein is widely expressed in kidney tissues (Abdelmegeed et al. 2017; Howard et al. 2003), while negative controls were used without primary antibodies and replaced with PBS. Please see the revised in lines 269-273 and 311-314.

References :

Abdelmegeed, M. A., Choi, Y., Ha, S.-K., & Song, B.-J. (2017). Cytochrome P450-2E1 is involved in aging-related kidney damage in mice through increased nitroxidative stress. Food and Chemical Toxicology, 109, 48–59. doi:10.1016/j.fct.2017.08.022

Howard L.A., Miksys S., Hoffmann E., Mash D., Tyndale R.F. Brain CYP2E1 is induced by nicotine and ethanol in rat and is higher in smokers and alcoholics. Br. J. Pharmacol. 2003;138:1376–1386. doi: 10.1038/sj.bjp.0705146.

Comment 6: Does the lower protein expression in muscle compared to the kidney hold any significance?

Response 6: We appreciate the reviewer's comments. The protein expression analysis only validated CYP2E1 protein expression in muscle tissues. We used kidneys as positive controls to test whether the CYP2E1 antibodies and our analysis methods were correct. The previous study reported that the CYP2E1 protein expression is often found in kidney tissues. Therefore, we tested whether CYP2E1 protein was also expressed in our sheep muscle samples.

Comment 7: What is the relationship between the data in Figure 4 and CYP2E1?

Response 7: Thanks for the question. Figure 4 shows the HE staining on the sheep muscle tissue. We carried out HE staining to see the morphology of sheep muscles based on the meat quality values, namely pH and tenderness. However, HE staining results (Figure 4) have no direct relationship to CYP2E1.

Comment 8: Additionally, why were liver tissues used for mRNA expression while phenotypes were measured in muscle tissues?

Response 8: Thanks for the comment. We analyzed mRNA expression on liver tissue because we referred to Listyarini et al. 2018 and Kubesova et al. 2019, which also used the same samples. Based on previous studies, CYP2E1 expression appeared frequently in liver samples. Gene expression will generally be distributed in all tissues. However, we tested on tissue with high expression values, namely liver tissues, for more accurate results. On the other hand, we measured meat phenotypes in muscle samples to measure meat quality values directly in muscle samples.

References :

Listyarini K, Jakaria, Uddin MJ, Sumantri C, Gunawan A. Association and expression of CYP2A6 and KIF12 genes related to lamb flavor and odor. Trop. Anim. Sci. J. 2018; 41(2): 100-107. https://doi.org/10.5398/tasj.2018.41.2.100. 

Kubesova A, Stastny K., Faldyna M, Sladek Z, Steinhauserova I, Borilova G, et al. mRNA Expression of CYP2E1, CYP2A19, CYP1A2, HSD3B, SULT1A1, and SULT2A1 genes in surgically castrated, immunologically castrated, entire male and female pigs and correlation with androstenone, skatole, indole, and Improvac-specific antibody levels. CJAS. 2019; 64 (2): 89–97. https://doi.org/10.17221/159/2018-CJAS.

Comments from Reviewer 2

Major comments:

L129: What is the definition of concentrate GTO3? Kindly specify in detail, avoiding from abbreviations. Furthermore, there is evidence to suggest that the JTTS oxen used in the slaughterhouse were obtained from multiple farms located in West Java, Indonesia, and were fed in group cages. Nonetheless, the descriptions given are extremely general. It is imperative to specify the types of forage and concentrated feed employed, as well as the duration of their administration.

Response: Thanks for the suggestion. GT-03 concentrate is a type of trade product from Indofeed. All rams were housed in group cages and given king grass forage (Pennisetum purpureophoides) ad libitum. In the afternoon, they were supplemented with GT-03 Indofeed concentrate in the amount of 400 g/head/day, while water was given ad libitum. We have added the explanation in the revised manuscript. Please see lines 136-139 in the revised manuscript with track changes.

L159-161: Cooked meat samples are typically employed for Warner-Bratzler Shear Force (WBSF) analysis. Kindly elaborate on the preparatory procedures preceding WBSF. In cases where WBSF was conducted on raw meat, as reported in certain studies, please specify this approach.

Response: Thanks for the comments. We have added the specific approach for WBSF analysis. For more explanation, please see the revised manuscript at lines 186-194.

L175: Please specify the column utilized for the fatty acid analysis and provide relevant properties. A similar situation is valid for cholesterol, flavor, and odor measurements.

Response: Thanks for the suggestion. We have added a column for fatty acid, cholesterol, flavor, and odor analysis for each procedure. Please see the revised manuscript on lines 207-209 for fatty acid, 226-228 for cholesterol, and flavor odor in lines 238-240.

L198-200: Which instrument did you employ for the mineral content analysis of the LD muscle, the ICP-OES or the AA spectrophotometer? Please provide specific details.

Response: We appreciate the reviewer's comments. Mineral content analysis was done using the atomic absorption spectrometry (AAS) method. For the detailed information, please see lines 245-249 in the revised manuscript.

L219-220: In your study, it was noted that JTTS rams were utilized, with a distribution of CC (n=33), CT (n=58), and TT (n=9) genotypes. Could you elaborate on the rationale and methodology employed for obtaining liver tissue samples from Garut composite sheep rams?

Response: Thanks for the suggestion. The quantitative real-time PCR analysis was carried out to observe the CYP2E1 gene expression based on differences in genotype. Liver tissue samples for gene expression analysis were selected from a total of 100 rams (CC (n=33), CT (n=58), and TT (n=9)). A total of 9 samples were selected representing three genotypes, CC, CT, and TT, each genotype consisting of 3 individuals. Three groups of rams were formed based on the TT, CT, and CC genotypes, which had high and low pH and tenderness values. Please see the revised manuscript with track changes for detailed changes (L69-273).

L261-262: In determining the grade of sheep groups, what specific criteria are utilized? Please share extra specific details about your methods.

Response: Thanks for the valuable comment and suggestion. The HE staining was carried out to see the morphology of lamb muscle based on the quality value of the meat, which, in this case, we determined based on the pH value and tenderness. We sorted individuals with high and low meat quality scores. We selected three rams representing each group for HE staining analysis. We have added the revision in the manuscript (L276).

L520-522: An apparent discrepancy is evident in either the materials and methods or in the discussions: While JTTS rams were indicated to have been used for slaughter in the materials and methods, there appears to be a contradiction in attributing the observed variances to breed differences.

Response: Thanks for the suggestions and comments. We analyzed CYP2E1 gene expression based on variations in the CYP2E1 genotype. We selected nine samples representing three genotypes, CC, CT, and TT, each genotype consisting of 3 individuals. However, we did not find variations in these three genotypes from the same breed, so the nine samples we chose were taken from the JTTS and GCS breeds. The analysis of genotype variations showed that the CT genotype had higher expression values than the CC and TT genotypes. The differences may also influence the higher expression in the sheep breeds. We have corrected the sentence marked in line 591-592.

Minor comments:

Please verify that abbreviations used throughout the text are appropriately placed.

Response: Thanks for the suggestion. We have corrected the abbreviations we use in the manuscript. For detailed changes, please see the revised manuscript with track changes.

L25: CYP2E1 appears in numerous places throughout the text, both in italics and in normal form. Each must be italicized for correction.

Response: We appreciate the reviewer's comments. We have rewritten all CYP2E1 according to the nomenclature requirements (https://academic.oup.com/molehr/pages/Gene_And_Protein_Nomenclature). When the text refers to the protein it shall not be italics. We use italics only when we refer to the gene or the mRNA. Please see the revised manuscript with track changes for detailed changes. 

L26: Please use “g. 50658168” instead of “50658168”.

Response: Thanks for the suggestion. We have corrected the sentence based on the suggestion. Please see the revised manuscript in line 26.

L28, L39, L148, L429: Please use “fatty acid” instead of “fat acid”.

Response: Thanks for the suggestion. We have used fatty acids in our manuscript. For detailed changes, please see the revised manuscript with track changes. 

L60, L61: Please use “grams/capita/year” instead of “grams per capita per year”.

Response: We appreciate the comments. We have rewritten "grams/capita/year." Please see the revised manuscript (L60, L61)

L313, L335: Please ensure that each abbreviation used in the table is accompanied by a corresponding footnote below the table

Response: Thanks for the suggestion. We have revised the table according to the reviewer's recommendation. Please see the revised manuscript with track changes for detailed changes (lines 373-374).

L340: Please use “fat” instead of “Fats” in Table 4.

Response: Thanks for this point. We have used fat in our table. Please see in Table 3 (L396)

L353: Please provide the section where lipid numbers are presented within parentheses.

Response: Thanks for the comment. We have presented the lipid number within parentheses. Please see in Table 3 (L396)

L386: Please use "potassium (K)" instead of "Kalium (K)."

Response: Thanks for the suggestion. We have used potassium (K) in our results. Please see the revised manuscript, especially in Table 3 (L396).

---

## [Decision Letter · Decision Letter 1]

12 Jul 2024

PONE-D-24-10259R1A comprehensive study of CYP2E1 and its role in carcass characteristics and chemical lamb meat quality in different Indonesian sheep breedsPLOS ONE

Dear Dr. Noor,

Thank you for submitting your manuscript to PLOS ONE. After careful consideration, we feel that it has merit but does not fully meet PLOS ONE’s publication criteria as it currently stands. Therefore, we invite you to submit a revised version of the manuscript that addresses the points raised during the review process.

We look forward to receiving your revised manuscript.

Kind regards,

Palash Mandal

Academic Editor

PLOS ONE

Additional Editor Comments:

Dear Authors,

Thank you for submitting your manuscript to PLOS ONE. After careful consideration, we feel that it has merit but does not fully meet PLOS ONE publication criteria as it currently stands. The shortcomings of this paper needs to be worked out before it can be considered for publication. Therefore, we invite you to resubmit a revised version of the manuscript that addresses the points raised during the review process.

For your guidance, the reviewers' comments are included below.

Thank you for giving us the opportunity to consider your work.

Specific concerns expressed during peer review were:

Reviewers' comments:

Reviewer's Responses to Questions

**Comments to the Author**

1. If the authors have adequately addressed your comments raised in a previous round of review and you feel that this manuscript is now acceptable for publication, you may indicate that here to bypass the “Comments to the Author” section, enter your conflict of interest statement in the “Confidential to Editor” section, and submit your "Accept" recommendation.

Reviewer #1: All comments have been addressed

Reviewer #3: (No Response)

2. Is the manuscript technically sound, and do the data support the conclusions?

Reviewer #1: Yes

Reviewer #3: Yes

3. Has the statistical analysis been performed appropriately and rigorously? 

Reviewer #1: Yes

Reviewer #3: Yes

4. Have the authors made all data underlying the findings in their manuscript fully available?

Reviewer #1: Yes

Reviewer #3: Yes

5. Is the manuscript presented in an intelligible fashion and written in standard English?

Reviewer #1: Yes

Reviewer #3: Yes

6. Review Comments to the Author

Reviewer #1: The author has responded to all the comments and performed necessary changes in the revised manuscript. The manuscript should be accepted for publication.

Reviewer #3: The paper entitled "A comprehensive study of CYP2E1 and its role in carcass characteristics and chemical lamb meat quality in different Indonesian sheep breeds" deals with an interesting topic with aim of evaluating the polymorphism of CYP2E1 gene and its role on the lamb meat quality of 7 Indonesian sheep breeds. Overall, the paper presented high quality of data, however, their discussion is lacking of scientific approach. Moreover, in the material and methods were not clear how much samples and replicates were used for each analyses, which is an important issue to be covered in order to demonstrate the strenghtness of the results obtained. Such an example, the data of polymorphism analysis were not clear on what number of samples have been obtained, on the contrary the real time PCR was performed on three animals for each species treated, I believe that is very low numbers of samples considering the importance of this analysis in the determining the role of the CYP2E1 in meat quality, in this context the main conclusion could not be supported by data. The main criticism is on the discussion which contains a large number of results, including p values, tables and figures references, so it was more a results section than discussion section, therefore, it need to be completely rewritten.

7. PLOS authors have the option to publish the peer review history of their article (what does this mean?). If published, this will include your full peer review and any attached files.

Reviewer #1: **Yes: **Jasim Muhammad Uddin, School of Veterinary Science, Murdoch University, Australia

Reviewer #3: No

---

## [Author Response · Author response to Decision Letter 1]

24 Jul 2024

Response to Reviewer's Comments

Reviewer #1: The author has responded to all the comments and performed necessary changes in the revised manuscript. The manuscript should be accepted for publication.

Response #1: We are grateful for your positive feedback and your consideration of our manuscript for publication.

Reviewer #3: The paper entitled "A comprehensive study of CYP2E1 and its role in carcass characteristics and chemical lamb meat quality in different Indonesian sheep breeds" deals with an interesting topic with aim of evaluating the polymorphism of CYP2E1 gene and its role on the lamb meat quality of 7 Indonesian sheep breeds. Overall, the paper presented high quality of data, however, their discussion is lacking of scientific approach. Moreover, in the material and methods were not clear how much samples and replicates were used for each analyses, which is an important issue to be covered in order to demonstrate the strenghtness of the results obtained. Such an example, the data of polymorphism analysis were not clear on what number of samples have been obtained, on the contrary the real time PCR was performed on three animals for each species treated; I believe that is very low numbers of samples considering the importance of this analysis in the determining the role of the CYP2E1 in meat quality, in this context the main conclusion could not be supported by data. The main criticism is on the discussion which contains a large number of results, including p values, tables and figures references, so it was more a results section than discussion section, therefore, it need to be completely rewritten.

Response #3: We appreciate your comments and suggestions, which reflect your deep understanding of the subject matter. We have explained the samples used in the material and method and have rewritten the discussion section according to your suggestion. In the real-time PCR analysis, we used a total of 9 samples representing three genotypes, CC, CT, and TT, each genotype consisting of three individuals. We acknowledge that the small sample size in our study may not fully represent the overall results. Therefore, we agree with your assessment that further validation is necessary to confirm the effect of CYP2E1 gene expression and polymorphisms in different sheep populations.

---

## [Decision Letter · Decision Letter 2]

28 Aug 2024

A comprehensive study of CYP2E1 and its role in carcass characteristics and chemical lamb meat quality in different Indonesian sheep breeds

PONE-D-24-10259R2

Dear Dr. Ronny Rachman Noor,

We’re pleased to inform you that your manuscript has been judged scientifically suitable for publication and will be formally accepted for publication once it meets all outstanding technical requirements.

Kind regards,

Palash Mandal

Academic Editor

PLOS ONE

---

## [Editor Report · Acceptance letter]

30 Aug 2024

PONE-D-24-10259R2 

PLOS ONE

Dear Dr. Noor, 

I'm pleased to inform you that your manuscript has been deemed suitable for publication in PLOS ONE. Congratulations! Your manuscript is now being handed over to our production team.

Kind regards, 

on behalf of

Prof. Palash Mandal 

Academic Editor

PLOS ONE